# Methicillin resistance *Staphylococcus aureus* nasal carriage and its associated factors among HIV patients attending art clinic at Dessie comprehensive specialized hospital, Dessie, North East Ethiopia

**Hussein Muhaba**[1], **Genet Molla Fenta**[2], **Daniel Gebretsadik**[2]*

**1** Amhara Public Health Institute, Dessie, Ethiopia, **2** Department of Medical Laboratory Science, College of Medicine and Health Sciences, Wollo University, Dessie, Ethiopia

* gebretsadikd@gmail.com

**Data Availability Statement:** The authors confirm that all data underlying the findings are fully

## Abstract

Globally the incidence of nosocomial infections and colonization due to methicillin resistant *Staphylococcus aureus* (MRSA) has become greater concern. The objective of the study was to determine the prevalence and associated factors of nasal carriage of MRSA with its antimicrobial susceptibility patter among HIV patients attending ART clinic. cross-sectional study was conducted from January 01 to May 30, 2020 at Dessie comprehensive specialized hospital, north east Ethiopia. A total of 206 HIV patients were recruited by applying systematic random sampling technique. Nasal specimen was collected from both anterior nares, and inoculated directly on mannitol salt agar, MacConkey, 5% blood agar. Screening of MRSA and methicillin susceptible *Staphylococcus aureus* (MSSA) strain was done by using cefoxitin antibiotic disc following modified Kirby-Bauer disc diffusion technique. Bivariable and multivariable logistic regression analyses were performed to assess the associated factors with *S. aureus* and MRSA. study participants were in the age range between 12 and 72 years and their mean (±SD) age was 41.52 (±11.2). The rate of *S. aureus* and MRSA colonization was 127/206 (61.7%) and 58/206 (28.2%), respectively. Having job close contact with human [AOR = 4.41; 95% CI = 1.5–13.02; p = 0.007], picking the nose [AOR = 4.38; 95% CI = 1.34–14.29; p = 0.014] and ART failure [AOR = 7.41; 95% CI = 2.08–26.41; p = 0.002] had statistically significant association with MRSA colonization. MRSA showed resistance for tetracycline (53.4%), erythromycin (84.5%), and trimethoprim-sulfamethoxazole (86.2%). Multi-Drug Resistance (MDR) was detected among 96.5% of MRSA and 20.3% of MSSA isolates. the rate of *S. aureus* and MRSA nasal colonization was high and it has associated with different factors. Understanding and managing MRSA among HIV patients is mandatory and stakeholders should find out the way how to decolonize the bacteria from nasal area.

available without restriction. All relevant data are within the manuscript.

**Funding:** The authors received no specific funding for this work.

**Competing interests:** The authors have declared that no competing interests exist.

## Introduction

Globally the incidence of nosocomial infections and colonization due to methicillin resistant *Staphylococcus aureus* (MRSA) particularly among healthcare facilities has become greater concern [1–3]. One study in Tanzania showed that patients can acquire MRSA while they are admitted in the hospital, demonstrating the possible transmission of healthcare acquired MRSA to any individuals who are working at or seeking medical services from such facilities [4]. This scenario ultimately attributed by genes that are identified among MRSA and Multi-drug resistant (MDR) isolates of the bacteria. Another attributing factor for the increased incidence of global nosocomial infection due to MRSA strain is the ability of the bacteria to form biofilm [1, 5, 6].

Colonization of *S. aureus* and MRSA strain were evident from different clinical specimens and from diverse study populations. A study conducted in Eritrea showed that a higher rate of colonization from pus specimen collected among different patients [7]. Other studies were conducted to demonstrate nasal colonization of *S. aureus* and MRSA from health care workers and hospital admitted patients [1, 3, 4]. Another studies also done to show nasal colonization rate of the bacteria with methicillin resistant pattern among HIV infected and non-infected individuals [8–12].

HIV-infected individuals are at greater risk for colonization with *S. aureus* and MRSA strains, this population group can carry the bacteria and led to consequent transmission to either susceptible or non-susceptible population. The prior colonization of *S. aureus* particularly, MRSA strain can also cause clinical infection among HIV patients [13]. This situation is supported by studies conducted in Ghana and Nepal in which HIV positive patients carry higher rate of *S. aureus* and MRSA in relative with HIV negative control groups [8, 14]. According to a Global wide estimation, there was a huge burden of MRSA colonization among HIV patients [15]. The burden of community associated MRSA among HIV patients was also higher than their counterparts [16].

Different studies across various countries, particularly in Africa, reported low to high nasal colonization of *S. aureus* and MRSA strain among HIV patients [9, 10, 17–19]. The rate of colonization was also affected by different factors such as antibiotic usage, CD4 count, history of hospitalization, viral load and so on [6, 17, 20, 21]. For example a systematic review and Meta analysis indicated that previous hospitalization, CD4 count and MRSA colonization history had 1.8, 1.79 and 6.3 times increased odds for MRSA colonization among HIV patients [22]. Another similar study also showed history of antibiotic use within 1 year had 3.89 times higher likelihood of acquiring MRSA colonization [12].

The isolates were also fostering wide ranged degree of resistance for antimicrobial agent [6, 18, 23]. As a result of colonization of MRSA either in the healthcare locations or in the community various forms of infection have been developed. In some situations HIV infection can be considered as the only factor for the colonization of MRSA [24], and in the course of time it was responsible for the occurrence of different infection including skin and soft tissue infections, pneumonia and bacteremia [13]. Hence the current study was conducted to determine the prevalence and associated factors of nasal carriage of MRSA with its antimicrobial susceptibility patter among HIV patients in Dessie comprehensive specialized hospital.

## Methods

### Ethics statement

The study was conducted after ethical clearance obtained from College of Medicine and Health Sciences, Wollo University, with a proposal number of CMHS 1140/2020. An official

cooperation letter was submitted to Dessie comprehensive specialized hospital and APHI-Dessie branch and obtained permission to conduct the study. The recruited study participants were involved after obtaining written informed consent for those whose age was greater than or equals to 18 and assent from their family or guardian for study participants whose age was below 18 years. The study participants were informed about the study process and its importance to an appropriate diagnostic procedure for the disease. Additionally, all the information obtained from the study subjects were coded to maintain confidentially. When the study participants are found to be positive for bacteria, they were informed by the health worker and receive proper treatment.

## Study area, design and period

A health facility based cross-sectional study was conducted at ART clinic of Dessie comprehensive specialized hospital in Dessie town from January 01 to May 30, 2020. The town is located in Eastern part of the Amhara Region and north central part of Ethiopia. It has a latitude and longitude of $11^0$8'N $39^o$38'E with an elevation between 2470 and 2,550 meters above sea level. The hospital serves for about seven million people living in the catchment areas. The hospital has six hundred inpatient beds and it also provides comprehensive Anti-Retroviral Therapy (ART) clients care service for 6,350 populations.

## Population

The source population was all people living with HIV who have enrolled at ART clinic. The study population was people living with HIV who have enrolled at ART clinic during the study period.

## Inclusion and exclusion criteria

HIV patients who were attending ART in the hospital and who were providing informed consent to participate in the study were included. For study participants whose age was less than 18 years written consent was obtained from the guardian/family of each study subjects. Whereas those who were complaining of frequent nasal bleeding and who had received antimicrobials prior to two weeks of sample collection were excluded from the study.

## Sample size determination and sampling technique

The sample size was computed using sample size determination formula for the estimation of the single population proportion, $n = \frac{(Z2/\alpha)^2 * P(1-P)}{d^2}$

Where; n = sample size, Z $\alpha$/2 = standardized normal distribution value at the 95% CI (95% CI two tailed, which is 1.96, d is desired precision /marginal error (5%) and P is pervious study prevalence, 16.8% [23]. In line with this consideration the calculated sample size was 214. Since the source population is 6,350; which is <10,000; the required minimum sample was obtained from the above estimate, it needs correction factors by making some adjustment (Cochran's formula).

Therefore, the final sample size of the study was 206. A systematic random sampling technique was applied to recruit study participants. The first sample was selected by the lottery method from the first K-unit of HIV patients, and the rest of potential participants were approached and screened during the recruitment period every K$^{th}$ HIV patient entering the ART Clinic until the achievement of the expected sample size within the given study period.

### Instruments of data collection method

Pretested structured questionnaire was used to gather primary data including socio-demographics information, behavioral and environmental factors for possible association with MRSA colonization. Information about clinical history and health conditions of each study subjects were retrieved from patients' registration chart. Microbiological data such as *S. aureus* colonization, AST profiles, and phenotypic characteristics identification for MRSA strain were collected.

### Data quality management

Every study participants have unique identification number. The data were entered in to EPI-data manager for consistency and cleaning. Completeness and consistency of data were checked by investigators before and after entry. Hence, all collected data were checked and cleaned in order to identify any inconsistencies and missing values and finally set up the necessary corrective measures.

### Sample collection

Nasal specimens from 206 study participants were collected. Specimens are collected from both anterior nares by inserting into the nostrils 1–2 cm inside, swabbing them in both clockwise and anticlockwise directions for 3–4 times of all eligible HIV patients. The collected specimens were transported by Amies transport media (Oxoid, UK) by applying a triple packaging system to the microbiology laboratory of Amhara Public Health Institute (APHI)-Dessie branch with the minimum delay of four hours for culture.

### Laboratory investigations

**Quality control.**   Reliability of the study findings was guaranteed through the implementation of standard quality control (QC) measures throughout the whole processes of the laboratory works. Sample collection, transportation and processing steps were performed following Standard operating procedures. All culture plates were prepared according to the manufacturers' instructions. Media preparation, inoculation, and culture were performed in strict aseptic conditions. The prepared media was checked for sterility and media performance testing of the appearance of pure growth of organisms by using ATCC control strain. The ATCC reference strain of *S. aureus* ATCC 25923 was used as a positive control organism for media capacity and antibiotic (drug) potency in the laboratory procedures [25]. In addition, *Escherichia coli* (ATCC 25922) strain was also used as a negative control. The thickness of MHA plates was maintained at 4 mm and the PH at 7.2–7.4. Microscope, incubator, centrifuge, refrigerator, water bath, autoclave and hot air oven were checked regularly to ensure the correct functioning of equipment for the reliability of results. The temperature for all equipment was monitored and recorded.

**Culture.**   The isolation and identification of *S. aureus* was carried out at APHI-Dessie branch. The swabs were inoculated directly on mannitol salt agar (Hi-Media), MacConkey (Hi-Media), 5% blood agar (Bio-Mark) and incubated at 35–37°C overnight. All positive cultures were identified by their characteristic appearance on media, Gram staining reactions and the pattern of biochemical profiles using standard procedures. *Staphylococcus aureus* were identified based on characteristic yellow colony surrounded by yellow zone on mannitol salt agar; β-hemolytic colonies with yellowish pigment on blood agar; no growth in MacConkey agar; Gram positive cocci singly in pair, in short chain or clusters in Gram stain; catalase positive, tube coagulase positive and mannitol fermentation in biochemical [25, 26].

**Antibiotic sensitivity testing.** It was performed on MHA (HI MEDIA) by modified Kirby-Bauer disk diffusion method as per the Clinical Laboratory and Standards Institute (CLSI) 2019 guidelines [25] and the diameter of zone of inhibition was measured and the obtained results were interpreted accordingly. The standard antibiotic discs (Liofilchem-Italy, HARDY Diagnosis-Santa Maria, USA)and its concentrations used as: Penicillinase-labile agents = Penicillin G (10 *units*); Penicillinase-stable agents = Cefoxitin (30 $\mu$g, Oxacillin 1$\mu$g); Macrolides = Erythromycin (15$\mu$g); Lincosamides = Clindamycin (2 $\mu$g); Folate pathway antagonists = Trimethoprim- Sulfamethoxazole 1.25/23.75$\mu$g;Tetracyclines = Tetracycline (30 $\mu$g); Phenicols = Chloramphenicol (30$\mu$g); Fluoroquinolones = Ciprofloxacin (5$\mu$g) and Aminoglycosides = Gentamycin (10$\mu$g).

**Screening of methicillin-resistant *S. aureus*.** In the absence of molecular techniques, the cefoxitin disc is the best predictor of methicillin resistance in *S. aureus* in antibiotic sensitivity testing [25]. Based on this evidence, the screening for MRSA and MSSA identification was done by using cefoxitin (30 $\mu$g) antibiotic disc following modified Kirby-Bauer disc diffusion technique with incubation period 24 hours in 35˚c. The *S. aureus* zone size of >22 mm around cefoxitin disk was confirmed as MSSA strain and ≤21 mm around cefoxitin disk was confirmed as MRSA strain as per the CLSI 2019 guidelines [25].

**Data processing, analysis method & interpretation.** Data entered in to EPI-data version 3.1 for consistency checks and data cleaning; and then exported to Statistical Package for Social Sciences (SPSS) software version 23 for analysis. Descriptive statistics, frequency and percentage were used to describe the study participants in relation to relevant variables. Bivariable and multivariable logistic regression analyses were performed to assess the associated factors with *S. aureus* and MRSA colonization. In all cases a 95% confidence interval and P–value ≤ 0.05 was considered as statistically significant. Finally, the results were explained in words and tables.

## Results

### Socio demographic characteristics

A total of 206 HIV positive clients who were attending the ART clinic were included in the study. Among them, female study participants account for 129 (62.6%), and 77 (37.4%) were male. The whole study participants were in the age range between 12 and 72 years and the mean (±SD) age of the study participants was 41.52 (±11.2). More than half of study participants were married, 121 (58.7%), 81 (39.3%) of the study participants had elementary level educational status. The majority of them, 166 (80.6%), were urban dwellers; one fourth of the study participants were employed (**Table 1**).

### Environmental, behavioral, clinical and health conditions characteristics

From a total of 206 study participants, 158 (76.7%) and 44 (21.4%) were on current WHO stage-I and II, respectively. Most 155 (75.2%) had an undetectable viral load and 19 (9.2%), had >1000 copies/mm$^3$ viral load counts. Most study participants were under first line antiretroviral regimen therapy 162 (78.6%) and followed by second line 41 (19.9%). The current body mass index (BMI) of 120 (58.3%) study participants was normal (between 18.50 Kg/m$^2$ and 24.99 Kg/m$^2$). About 21.4% of the patients had a history of ART failure and 19.4% had a history of hospital admission within the past 12 months. Among 206 participants, 150 (72.8%) had HAART initiation time greater than a five year period, living longer with HIV carrier and the rest, 56 (27.2%) living up to five years long (**Table 2**).

**Table 1. Socio demographic characteristics of study participants at ART clinic in Dessie comprehensive specialized hospital from January to May 2020 (n = 206).**

| Variables | Category | Frequency | (%) |
|---|---|---|---|
| Sex | Male | 77 | 37.4 |
| | Female | 129 | 62.6 |
| Age | <20 | 4 | 1.9 |
| | 20–29 | 16 | 7.8 |
| | 30–39 | 73 | 35.4 |
| | 40–49 | 55 | 26.7 |
| | 50–59 | 44 | 24.4 |
| | >60 | 14 | 6.8 |
| Education | Illiterate | 32 | 15.5 |
| | Elementary | 81 | 39.3 |
| | High school | 67 | 32.5 |
| | College and above | 26 | 12.6 |
| Marital Status | Single | 25 | 12.1 |
| | Married | 121 | 58.7 |
| | Divorced | 22 | 10.7 |
| | Widowed | 38 | 18.4 |
| Residence | Rural | 40 | 19.4 |
| | Urban | 166 | 80.4 |
| Occupation | Employed | 53 | 25.7 |
| | Merchant | 45 | 21.8 |
| | Farmer | 19 | 9.2 |
| | Housewife | 49 | 23.8 |
| | Daily worker | 26 | 12.6 |
| | Others | 14 | 6.8 |
| Number of family members | One | 19 | 9.2 |
| | Two | 42 | 20.4 |
| | Three | 53 | 25.7 |
| | Four | 46 | 22.3 |
| | Five | 23 | 11.2 |
| | ≥six | 23 | 11.2 |

## Prevalence of bacterial growth on nasal area

From a total of 206 nasal specimens, 187 (90.8%) had bacterial growth while19 (9.2%) did not provide any bacterial growth. A total of 16 bacterial species were identified and out of these, 127 (61.7%) were *S. aureus*, followed by *CoNS* 43 (20.9%) and *E.coli*27 (13.1%) (**Table 3**).

From the total of 206 nasal swab cultures, the rate of MSSA colonization was 33.5% (69) while MRSA was 28.2% (58). The proportion of MRSA strain among female study participants was higher than males (18.4% vs. 9.7%); the highest and lowest rate of MRSA was also indicated among study participants who were categorized at 30–49 and greater than 60 years. Urban inhabitants and married study participants also had higher rates of MRSA and *S. aureus* nasal colonization (**Table 4**).

## Analysis of associated factors for *S. aureus* and MRSA colonization

Independent variables were analyzed using a bivariable and multivariable logistic regression model for the possible association with nasal colonization of *S. aureus* and MRSA among ART

**Table 2. Environmental, behavioral, clinical history and health conditions of nasal colonization of MRSA suspected ART clients attending at Dessie comprehensive specialized hospital ART clinic from January to May 2020 (n = 206).**

| Variables | Category | Frequency | (%) | Variables | Category | Frequency | (%) |
|---|---|---|---|---|---|---|---|
| Job-Close Contact | Close contact | 148 | (71.8) | WBC count base line | Low | 53 | (25.7) |
| | No contact | 58 | (28.2) | | Normal | 131 | (63.6) |
| Long Nails | Yes | 22 | (10.7) | | High | 22 | (10.7) |
| | No | 184 | (89.3) | WBC count most recent | Low | 17 | (8.3) |
| Current smoker | Yes | 15 | (7.3) | | Normal | 170 | (82.5) |
| | No | 191 | (92.7) | | High | 19 | (9.2) |
| Current drunker | Yes | 23 | (11.2) | Hospitalized[1] | Yes | 40 | (19.4) |
| | No | 183 | (88.8) | | No | 166 | (80.6) |
| Current chat chewer | Yes | 33 | (16.0) | Hospitalized[2] | Yes | 12 | (5.8) |
| | No | 173 | (84.0) | | No | 194 | (94.2) |
| HAART Initiation time | Within six months | 9 | (4.4) | Antibiotics[3] | Yes | 99 | (48.1) |
| | 7–12 months | 2 | (1.0) | | No | 107 | (51.9) |
| | 1–5 yrs | 45 | (21.8) | Prophylaxis TMP.SMX | Previous | 115 | (55.8) |
| | >5 yrs | 150 | (72.8) | | Current | 78 | (37.9) |
| Base line WHO clinical stage | Stage-I | 4 | (1.9) | | No | 13 | (6.3) |
| | Stage-II | 52 | (25.2) | Current BMI | Under weight | 13 | (6.3) |
| Stage-III | 126 | (56.3) | Medium | | 31 | (15.0) | |
| Stage-IV | 34 | (16.5) | Normal | | 120 | (58.3) | |
| Current WHO | Stage-I | 158 | (76.7) | | Over weight | 39 | (18.9) |
| clinical stage | Stage-II | 44 | (21.4) | | Obesity-class-I | 3 | (1.5) |
| | Stage-III | 3 | (1.5) | Co-infection | Yes | 43 | (20.9) |
| | Stage-IV | 1 | (0.5) | | No | 163 | (79.1) |
| CD4 count base line | 0–200 | 129 | (62.6) | ART failure | Yes | 44 | (21.4) |
| | 201–350 | 58 | (28.2) | | No | 162 | (78.6) |
| | 351–500 | 13 | (6.3) | Viral load count base line | Not done | 8 | (3.9) |
| | >500 | 6 | (2.9) | | Undetected | 153 | (74.3) |
| CD4 count most recent | 0–200 | 10 | (4.9) | | <150 | 13 | (6.3) |
| | 201–350 | 28 | (13.6) | | 151–1000 | 1 | (0.5) |
| | 351–500 | 60 | (29.1) | | >1000 | 31 | (15.0) |
| | >500 | 108 | (52.4) | Viral load count most recent | Not done | 8 | (3.9) |
| ART regimen | First Line | 162 | (78.6) | | Undetected | 155 | (75.2) |
| | Second line | 41 | (19.9) | | <150 | 22 | (10.7) |
| | Third line | 3 | (1.5) | | 151–1000 | 2 | (1.0) |
| | | | | | >1000 | 19 | (9.2) |

Key: HAART: Highly Active Antiretroviral Therapy

[1]hospitalization within previous 12 month

[2]house hold member hospitalization within previous 12 months

[3]Antibiotics within previous 6 months, BMI: Body mass index (weight in kilograms divided by the square of the height in meters (kg/m2))

patients. In both cases, each variable that indicated P-value < 0.25 during the bivariable regression analysis was further analyzed using a multivariable analysis model. In the multivariable analysis model, there were five variables that had association with the colonization of *S. aureus*. In reference to college and above educational status study participants who are illiterate had more than five times the likelihood of *S. aureus* colonization [AOR = 5.22, 95%CI = 1.29–21.1, P-value = 0.02]. Study participants who had Job close contact with humans had 6.5 times the odds of colonizing by *S. aureus* [AOR = 6.52, 95%CI = 2.78–15.31, P-value ≤ 0.01] (**Table 5**).

**Table 3. Bacterial isolate on nasal area among suspected ART clients attending at Dessie comprehensive specialized hospital ART clinic from January to May 2020 (n = 206).**

| Isolated Micro-organisms | | No of strains (%) |
|---|---|---|
| *S. aureus* | | *127 (61.7)* |
| CoNS | | 43 (20.9) |
| *Streptococcus pneumonia* | | 2 (1.0) |
| *Streptococcus agalactiae* | | 1 (0.5) |
| *Enterococcus species* | | 1 (0.5) |
| E. coli | | 27 (13.1) |
| *Enterobacter species* | | 1 (0.5) |
| *Pseudomonas species* | | 6 (2.9) |
| *Acinetobacter species* | | 2 (1.0) |
| *Klebsiella pneumonia* | | 4 (1.9) |
| *Klebsiella ozaenae* | | 6 (2.9) |
| *Klebsiella oxytoca* | | 2 (1.0) |
| *Klebsiella rhinoscleroma* | | 2 (1.0) |
| *Proteus species* | | 2 (1.0) |
| *Morganella morganii* | | 1 (0.5) |
| *Serratia species* | | 1 (0.5) |
| Growth | Yes | 187 (90.8) |
| | No | 19 (9.2) |

In bivariable analysis, MRSA colonization rate showed association with ART failure, close contact job, current chat chewing and smoking, having CD4 count <200 cells/mm$^3$ and viral load counts >1000 copies/mm$^3$. Moreover, in multivariate analysis, having job close contact with humans [AOR = 4.41; 95% CI = 1.5–13.02; p = 0.007], Picking the nose [AOR = 4.38; 95% CI = 1.34–14.29; p = 0.014] and ART failure [AOR = 7.41; 95% CI = 2.08–26.41; p = 0.002] had statistically significant association with nasal colonization of MRSA (**Table 6**).

## Antibiotic susceptibility tests for *S. aureus* and MRSA

Overall, AST for *S. aureus* isolates against the ten most commonly used antibiotics were performed based on 2019 CLSI guideline. The strains demonstrated as various rate of resistance against ciprofloxacin (12.6%), tetracycline (37%), erythromycin (41.7%), and trimethoprim-sulfamethoxazole (62.2%). The sensitivity of the isolate against oxacillin (54.3%), chloramphenicol (91.3%), clindamycin (95.3%) and gentamycin (97.6%) were also demonstrated. The whole isolates of *S. aureus* (MSSA and MRSA) showed full resistance to penicillin (**Table 7**).

## Drug resistance pattern and multi-drug resistance

Overall, *S. aureus* isolates showed drug resistance in various patterns: R1 = 34 (26.8%), R2 = 23 (18.1%), R3 = 11 (8.7%), R4 = 27 (21.3%) and R>5 = 32 (25.2%). None of the isolates showed full sensitivity to the tested antibiotics. Multidrug resistances (MDR) were extrapolated as resistance to three or more groups of antibiotics were detected in nasal site with the prevalence of *S. aureus* isolates was found 55.2% (>R3 = 70) while MSSA and MRSA isolates were (>R3 = 14 (20.3%)) and (>R3 = 56 (96.5%)), respectively (**Table 8**).

## Discussions

In the present study, the rate of *S. auras* colonization was found to be 127 (61.7%) and the rate of MRSA was 58 (28.2%). The rate of *S. aureus* in the present study was comparable (50.5%)

**Table 4. Socio demographic distribution of *S. aureus*, MSSA and MRSA on nasal area among suspected ART clients attending at Dessie comprehensive specialized hospital ART clinic from January to May 2020 (n = 206).**

| Variable | Category | No | *S. aureus* colonization | | MRSA colonization | |
|---|---|---|---|---|---|---|
| | | | Yes: n(%) | No: n (%) | Yes: n (%) | No: n (%) |
| Sex | Male | 77 | *46 (22.3)* | 31(15.0) | 20 (9.7) | 26 (12.6) |
| | Female | 129 | *81 (39.3)* | 48 (23.3) | 38 (18.4) | 43 (20.9) |
| Age | <30 | 20 | *13 (6.3)* | 7 (3.4) | 5 (2.43) | 8 (3.9) |
| | 30–49 | 128 | *86 (41.75)* | 42 (20.4) | 41 (19.9) | 45 (21.8) |
| | 50–59 | 44 | *21(10.2)* | 23 (11.2) | 10 (4.9) | 11 (5.3) |
| | >60 | 14 | *7 (3.4)* | 7 (3.4) | 2 (1.0) | 5 (2.4) |
| Education | Illiterate | 32 | 22 (10.7) | 10 (4.9) | 5 (2.4) | 17 (8.3) |
| | Elementary | 81 | 53 (25.7) | 28 (13.6) | 27 (13.1) | 26 (12.6) |
| | High school | 67 | 42 (20.4) | 25 (12.1) | 20 (9.7) | 22 (10.7) |
| | Post education | 26 | 10 (4.9) | 16 (7.8) | 6 (2.9) | 4 (1.9) |
| Marital status | Single | 25 | 20 (9.7) | 5 (2.4) | 11 (5.3) | 9 (4.4) |
| | Married | 121 | 68 (33.0) | 53 (25.7) | 26 (12.6) | 42 (20.4) |
| | Divorced | 22 | 17 (8.3) | 5 (2.4) | 8 (3.9) | 9 (4.4) |
| | Widowed | 38 | 22 (10.7) | 16 (7.8) | 13 (6.3) | 9 (4.4) |
| Residence | Rural | 40 | 26 (12.6) | 14 (6.8) | 9 (4.4) | 17 (8.3) |
| | Urban | 166 | 101 (49.0) | 65 (31.6) | 49 (23.8) | 52 (25.2) |
| Occupation | Employee | 53 | 28 (13.6) | 25 (12.1) | 17(8.3) | 11 (5.3) |
| | Merchant | 45 | 32 (15.5) | 13 (6.3) | 14 (6.8) | 18 (8.7) |
| | Farmer | 19 | 12 (5.8) | 7 (3.4) | 4 (1.9) | 8 (3.9) |
| | Housewife | 49 | 25 (12.1) | 24 (11.7) | 11 (5.3) | 14 (6.8) |
| | Daily worker | 26 | 20 (9.7) | 6 (2.9) | 8 (3.9) | 12 (5.8) |
| | Others | 14 | 10 (4.9) | 4 (1.9) | 4 (2.0) | 6 (2.9) |
| | | | 127 (61.7) | 79 (38.3) | 58 (28.2) | 69 (33.5) |

Key: Others; Pension, Student, Homeless and beggar, Employee; Government and private employee.

with a study conducted in Ethiopia among pediatric HIV patients [2] but another study in Ethiopia reported a lower prevalence of *S. aureus* colonization among adult HIV patients (39.7%) [6]. Still, a very low rate was also reported in another similar studies in Ethiopia, other African countries and China [10, 17, 18, 27]. In contrary to the present study almost two fold higher MRSA colonization was reported in a study conducted in East Africa [28]. The discrepancy across studies might be partly due to difference in nature of population, specimen used, Methicillin-resistant coagulase-negative *Staphylococci* colonization and partly due to study period variation.

Two meta-analysis studies [15, 29] indicated a very low rate of MRSA colonization (6.9% and 7%) among HIV patients in relation with the present study. In Ethiopia a systematic review and meta-analysis on nasal colonization of *S. aureus* and MRSA among the different population group was done and the pooled prevalence of both colonization were found to be 30.9% and 10.94%, respectively which is still lower than the present study [30]. Comparable prevalence of MRSA colonization was reported from another systematic review and meta-analysis work in Ethiopia [31].

In reference to college and above educational status, study participants who are illiterate had more than five times the likelihood of *S. aureus* colonization of the nasal area. This probably indicates that a literate population may have good knowledge about the importance of sanitation to prevent other infections. Another statistically associated factor was having job-close

**Table 5. Bivariable and multivariable analysis for nasal colonization of *S. aureus* among ART clients attending at Dessie comprehensive specialized hospital ART clinic from January to May 2020.**

| Variable | Category | Bivariable analysis | | | | Multivariable analysis | | | |
|---|---|---|---|---|---|---|---|---|---|
| | | COR | (95% CI) | | P-value | AOR | (95% CI) | | P-value |
| | | | Lower | upper | | | Lower | upper | |
| Sex | Male | Ref. | | | | | | | |
| | Female | 1.14 | 0.64 | 2.03 | 0.663 | NA | - | | - |
| Age | | | | | 0.109 | | | | 0.075 |
| | < 30 | 1.86 | 0.46 | 7.48 | 0.384 | 4.45 | 0.46 | 43.12 | 0.197 |
| | 30–49 | 2.05 | 0.67 | 6.22 | 0.206 | 1.21 | 0.24 | 6.07 | 0.817 |
| | 50–59 | 0.91 | 0.27 | 3.04 | 0.882 | 0.46 | 0.08 | 2.48 | 0.364 |
| | > 60 | Ref. | | | | Ref. | | | |
| Education | | | | | 0.083 | | | | 0.1 |
| | Illiterate | 3.52 | 1.19 | 10.4 | 0.023 | 5.22 | 1.29 | 21.1 | 0.02* |
| | Elementary | 3.03 | 1.22 | 7.55 | 0.017 | 2.8 | 0.87 | 9.01 | 0.084 |
| | High school | 2.7 | 1.06 | 6.83 | 0.038 | 1.76 | 0.52 | 5.95 | 0.36 |
| | College and above | Ref | | | | Ref. | | | |
| Residence | Rural | 1.2 | 0.58 | 2.46 | .0.628 | NA | | | |
| | Urban | Ref. | | | | | | | |
| Marital status | | | | | 0.066 | | | | 0.596 |
| | Single | Ref. | - | | - | Ref. | - | | - |
| | Married | 0.32 | 0.11 | 0.91 | 0.033 | 1.07 | 0.18 | 6.31 | 0.94 |
| | Divorced | 0.85 | 0.21 | 3.44 | 0.82 | 2.56 | 0.29 | 22.7 | 0.4 |
| | Widowed | 0.34 | 0.11 | 1.11 | 0.074 | 1.43 | 0.21 | 9.96 | 0.72 |
| Job-Close Contact with human | Yes | 4.45 | 2.34 | 8.46 | ≤0.01 | 6.52 | 2.78 | 15.31 | ≤0.01* |
| | No | Ref. | - | | - | Ref. | - | | - |
| Current smoking | Yes | 9.66 | 1.24 | 75.0 | 0.03 | 2.3 | 0.17 | 30.92 | 0.53 |
| | No | Ref. | | | Ref. | | | | |
| Current drunker | Yes | 1.48 | 0.58 | 3.78 | 0.41 | NA | | | |
| | No | Ref. | | | | | | | |
| Current chat chewer | Yes | 3.29 | 1.29 | 8.37 | 0.013 | 2.27 | 0.59 | 8.71 | 0.23 |
| | No | Ref. | | | | Ref. | | | |
| CD4 count most recent | | | | | 0.086 | | | | 0.1 |
| | 0–200 | 7.48 | 0.92 | 61.1 | 0.061 | 18.96 | 1.44 | 248.8 | 0.025* |
| | 201–350 | 1.28 | 0.55 | 3 | 0.564 | 0.998 | 0.3 | 3.3 | 0.997 |
| | 351–500 | 1.94 | 0.99 | 3.79 | 0.053 | 1.76 | 0.74 | 4.14 | 0.198 |
| | >500 | Ref. | | | | Ref. | | | |
| Initiation of HAART | ≤ 5 years | 0.63 | 0.34 | 1.18 | 0.147 | 0.39 | 0.16 | 0.96 | 0.041* |
| | >5 years | Ref. | | | | Ref. | | | |
| Line of ARV therapy regimen | First | Ref. | | | | Ref. | | | |
| | Second or third | 8.62 | 2.95 | 25.2 | ≤0.01 | 7.43 | 2.03 | 27.24 | 0.002* |
| Current BMI | | | | | 0.304 | | | | 0.197 |
| | <18.5 | 5.3 | 0.44 | 64.35 | 0.19 | 1.85 | 0.13 | 26.54 | 0.65 |
| | 18.5–24.99 | 2.96 | 0.26 | 33.54 | 0.38 | 0.64 | 0.05 | 8.84 | 0.74 |
| | 25–29.99 | 2.71 | 0.23 | 32.34 | 0.43 | 1.03 | 0.07 | 15.18 | 0.98 |
| | ≥30 | Ref. | | | | Ref. | | | |
| Hospitalized[1] | Yes | 1.58 | 0.75 | 3.32 | 0.23 | 1.22 | 0.43 | 3.44 | 0.712 |
| | No | Ref. | – | | | Ref. | – | | |

(*Continued*)

**Table 5.** (Continued)

| Variable | Category | Bivariable analysis | | | | Multivariable analysis | | | |
|---|---|---|---|---|---|---|---|---|---|
| | | COR | (95% CI) | | P-value | AOR | (95% CI) | | P-value |
| | | | Lower | upper | | | Lower | upper | |
| Hospitalized[2] | Yes | 1.26 | 0.37 | 4.33 | 0.713 | NA | | | |
| | No | Ref. | | | | | | | |
| Co-infection | Yes | 1.57 | 0.76 | 3.23 | 0.22 | 0.95 | 0.35 | 2.6 | 0.927 |
| | No | Ref. | | | | Ref. | | | |

Key: Employed: both Governmental and private CI: Confidence interval; COR: Crude odds ratio; AOR: Adjusted odds ratio

[1]within previous 12 months

[2]household member hospitalized,; Ref: reference category.

contact with humans. In this regard, those study participants who had job close contact with humans showed a 6.5 times higher likelihood of colonization with *S. aureus* than persons who had not (P ≤ 0.01). This positive association between job close contact with humans and *S. aureus* colonization of the nasal area among HIV patients was supported by a study conducted in Nepal [8].

The role of T-cells is very huge in the clearance of *S. aureus* colonization from the nasal area where it is the major reservoir of the bacteria [32]. In the present study most recent CD4 count <200 (AOR = 18.96, 95%CI = 1.44 to 248.8, P = 0.025) and stayed on HAART for less than or equals to 5 years (AOR = 0.39, 95%CI = 0.16to 0.96, P = 0.041) had strong association with *S. aureus* colonization. Similar to the present study, the finding of a study conducted in Mekelle, Ethiopia also revealed that the CD4 count <200 cell/mm$^3$ has a positive association with nasal colonization of the bacteria [17]. Among HIV infected individuals with functional defects in CD4 cell responses, *S. aureus* can demonstrate infection which is beyond colonization [33, 34].

In multivariate analysis having job close contact with humans and the frequent habit of picking the nose by their hand and nail had 4.41 and 4.38 times higher odds of colonization with MRSA in comparison with their counterparts. Even though limited studies try to demonstrate the association between job close contact with human and MRSA colonization, the results of two studies showed agreement with the current study [8, 27]. One study assessed the relationship between MRSA colonization and frequent nose picking habit of HIV patients and the result of this study concluded that there is no any statistical association [19] which is not in agreement with the present study. A systematic review and meta-analysis in Ethiopia suggested maintaining hand hygiene, promoting environmental sanitation and health education as a means of reducing the colonization of MRSA [31]. Study participants may have a habit of sneezing into their hands, the pathogen may also transfer to other individuals while contacting with the colonized hands and then he/she could have the habit of picking their nose. This might be a possible scenario that could happen among the participants of the present study.

In the present study, participants who were on second or third line ARV therapy (first line ART failure) had demonstrated statistically significant association with the prevalence of *S. aureus* [AOR = 7.43, 95%CI = 2.03 to 27.24, P = 0.002] and MRSA [AOR = 7.41, 95%CI = 2.08 to 26.41, P = 0.002] colonization. First line ARV failure was reported in studies conducted in Addis Ababa and Gondar [35–37]. Presence of opportunistic infection, no formal education, low CD4 count and patients having respiratory infection like tuberculosis are important predictors of treatment failure among HIV patients [35, 36, 38, 39]. *Staphylococcus aureus* is normal human bacterial flora and is considered as a cause of a wide range of opportunistic

**Table 6. Bivariable and multivariable analysis of MRSA colonization with associated factors among ART clients attending at Dessie comprehensive specialized hospital ART clinic from January to May 2020.**

| Variable | Category | No | Bivariable analysis | | | | Multivariable analysis | | | |
|---|---|---|---|---|---|---|---|---|---|---|
| | | | COR | (95% CI) | | P-value | AOR | (95% CI) | | P-value |
| | | | | Lower | upper | | | Lower | upper | |
| Sex | Male | 20 | Ref. | | | | | | | |
| | Female | 38 | 0.84 | 0.44 | 1.58 | 0.591 | | | | |
| Age | | | | | | 0.404 | | | | 0.87 |
| | < 30 | 5 | 2.00 | 0.33 | 12.18 | 0.452 | 1.05 | 0.07 | 14.9 | 0.973 |
| | 30–49 | 41 | 2.83 | 0.61 | 13.22 | 0.187 | 1.53 | 0.26 | 9.14 | 0.643 |
| | 50–59 | 10 | 1.76 | 0.34 | 9.23 | 0.501 | 1.08 | 0.17 | 7.1 | 0.932 |
| | >59 | 2 | Ref. | - | - | - | Ref. | - | - | - |
| Education | | | | | | 0.28 | | NA | | |
| | Illiterate | 5 | 0.62 | 0.16 | 2.31 | 0.474 | | | | |
| | Elementary | 27 | 1.67 | 0.6 | 4.63 | 0.328 | | | | |
| | High school | 20 | 1.42 | 0.5 | 4.06 | 0.515 | | | | |
| | College and above | 6 | Ref. | - | | - | | | | |
| Residence | Rural | 9 | 0.69 | 0.31 | 1.56 | 0.38 | | NA | | |
| | Urban | 49 | Ref. | - | | - | | | | |
| Marital status | | | | | | 0.073 | | | | 0.386 |
| | Single | 11 | Ref. | - | | - | | | | - |
| | Married | 26 | 0.35 | 0.14 | 0.86 | 0.022 | 0.48 | 0.09 | 2.5 | 0.387 |
| | Divorced | 8 | 0.73 | 0.22 | 2.35 | 0.595 | 0.73 | 0.11 | 5.01 | 0.751 |
| | Widowed | 13 | 0.66 | 0.24 | 1.86 | 0.435 | 1.10 | 0.18 | 6.93 | 0.918 |
| Job-Close Contact | Close contact | 52 | 4.69 | 1.89 | 11.66 | 0.001 | 4.41 | 1.5 | 13.02 | 0.007* |
| | No contact | 6 | Ref. | - | | - | Ref. | - | | - |
| Keeping Long Nails | Yes | 9 | 1.91 | 0.77 | 4.74 | 0.165 | 1.12 | 0.28 | 4.5 | 0.877 |
| | No | 49 | Ref. | - | | - | Ref. | - | | - |
| Picking the nose | Always | 11 | 2.24 | 0.95 | 5.28 | 0.065 | 4.38 | 1.34 | 14.29 | 0.014* |
| | Sometimes | 47 | Ref. | - | | - | Ref. | - | | - |
| Current smoking | Yes | 9 | 4.35 | 1.47 | 12.84 | 0.008 | 0.61 | 0.08 | 4.69 | 0.632 |
| | No | 49 | Ref. | - | | - | Ref. | - | | - |
| Current drunker | Yes | 7 | 1.13 | 0.44 | 2.91 | 0.797 | | NA | | |
| | No | 51 | Ref. | - | | - | | | | |
| Current chat chewer | Yes | 15 | 2.52 | 1.17 | 5.42 | 0.018 | 0.94 | 0.24 | 3.62 | 0.925 |
| | No | 43 | Ref. | - | | - | Ref. | | | |
| Viral load count most recent | | | | | | ≤0.01 | | | | 0.851 |
| | Undetected | 29 | Ref. | - | | - | Ref. | | | - |
| | <150 | 11 | 4.34 | 1.72 | 10.99 | 0.002 | 1.82 | 0.48 | 6.89 | 0.379 |
| | 151–1000 | 1 | 4.34 | 0.26 | 71.52 | 0.304 | 1.33 | 0.03 | 51.73 | 0.88 |
| | >1000 | 13 | 9.41 | 3.3 | 26.85 | ≤0.01 | 1.3 | 0.23 | 7.29 | 0.762 |
| | Not done | 4 | | | | | | | | |
| CD4 count most recent | | | | | | 0.168 | | | | 0.882 |
| | 0–200 | 5 | 3.5 | 0.94 | 13.1 | 0.063 | 1.68 | 0.16 | 17.26 | 0.661 |
| | 201–350 | 9 | 1.66 | 0.66 | 4.13 | 0.278 | 1.22 | 0.35 | 4.25 | 0.76 |
| | 351–500 | 20 | 1.75 | 0.87 | 3.53 | 0.119 | 1.42 | 0.57 | 3.53 | 0.454 |
| | >500 | 24 | Ref. | - | | - | Ref | - | | - |
| Initiation of HAART | < = 5 years | 15 | 0.91 | 0.46 | 1.81 | 0.789 | | NA | | |
| | >5yrs | 43 | Ref. | - | | - | | | | |

*(Continued)*

**Table 6.** (Continued)

| Variable | Category | No | Bivariable analysis | | | | Multivariable analysis | | | |
|---|---|---|---|---|---|---|---|---|---|---|
| | | | COR | (95% CI) | | P-value | AOR | (95% CI) | | P-value |
| | | | | Lower | upper | | | Lower | upper | |
| Line of ARV therapy regimen | First | 30 | Ref. | - | | - | Ref. | - | | - |
| | Second and Third | 28 | 7.7 | 3.71 | 16 | ≤0.01 | 7.41 | 2.08 | 26.41 | 0.002* |
| Hospitalized[1] | Yes | 16 | 1.97 | 0.96 | 4.06 | 0.066 | 1.44 | 0.5 | 4.09 | 0.498 |
| | No | 42 | Ref. | - | | - | Ref. | - | | - |
| Hospitalized[2] | Yes | 3 | 0.84 | 0.22 | 3.23 | 0.802 | | NA | | |
| | No | 55 | Ref. | - | | - | | | | |
| Co-infection | Yes | 16 | 1.71 | 0.84 | 3.48 | 0.14 | 0.7 | 0.24 | 1.99 | 0.499 |
| | No | 42 | Ref. | - | | - | Ref. | - | | - |

Key: MRSA: methicillin-resistant S. aureus; CI: Confidence interval; COR: Crude odds ratio; AOR: Adjusted odds ratio

[1]within previous 12 months

[2]household member within previous 6 months, Ref: reference category.

infections [40]. The risk of colonization and diseases caused by *S. aureus* are increased among HIV infected individuals [8].

In the present study the rate of nasal colonization of *S. aureus* and MRSA among females were 81 (39.3%) and 38 (18.4%), respectively which is higher than male study participants. Similarly, a study conducted in Ethiopia also indicated that the rate of MRSA colonization was higher among female study participants [6]. Higher rate of *S. aureus* 86(41.75)and MRSA 41 (19.9) nasal colonization was also indicated among study participants who are in the age category of 30 to 49 years. According to a study conducted in Nigeria [41] the highest MRSA carriage rate was demonstrated among study participants who were in the age category between 31 and 40 years. In this study age was also indicated as a significantly associated factor which is not in agreement with the current study. Similar to other studies CD4 count of the study participants and use of co-trimoxazole in the current study had no statistical significant association with MRSA colonization [6, 27, 42]. History of hospitalization and hospitalization of once family member are considered as an associated factor for the nasal carriage rate of *S. aureus* in another study in Ethiopia [17]. But these factors did not statistically associated with the MRSA and *S. aureus* colonization in the present study.

**Table 7. Antibiotic susceptibility pattern of MSSA and MRSA suspected nasal area among ART clients attending at Dessie comprehensive specialized hospital ART clinic from January to May 2020.**

| Isolated Micro-Organisms | | Antimicrobial agents, n (%) | | | | | | | | | |
|---|---|---|---|---|---|---|---|---|---|---|---|
| | | Ox | E | CL | SXM | TET | CAF | CIP | GEN | PEN | CXT |
| MSSA (n = 69) | S | 69 (54.3%) | 65 (94.2%) | 69 (100%) | 40 (58.0%) | 53 (76.8%) | 66 (95.7%) | 64 (92.8%) | 69 (100%) | 0 | 69 (100%) |
| | R | 0 | 4 (5.8%) | 0 | 29 (42.0%) | 16 (23.2%) | 3 (4.3%) | 5 (7.2%) | 0 | 69 (100%) | 0 |
| MRSA (n = 58) | S | 0 | 9 (15.5%) | 52 (89.7%) | 8 (13.8%) | 27 (46.6%) | 50 (86.2%) | 47 (81.0%) | 55 (94.8%) | 0 | 58 (100%)0 |
| | R | 58 (45.7) | 49 (84.5) | 6 (10.3) | 50 (86.2) | | | | | | |
| Total (n = 127) | S | 69 (54.3%) | 74 (58.3%) | 121 (95.3%) | 48 (37.8%) | 80 (63.0%) | 116 (91.3%) | 111 (87.4%) | 124 (97.6%) | 0 | 69 (54.3%) |
| *S. aureus* | R | 58 (45.7%) | 53 (41.7%) | 6 (4.7%) | 79 (62.2%) | 47 (37.0%) | 11 (8.7%) | 16 (12.6%) | 3 (2.4%) | 127 (100%) | 58 (45.7%) |

Key: CXT = Cefoxitin, Ox = Oxacillin, E = Erythromycin, CL = Clindamycin, SXT-SXM = Trimethoprim- Sulfamethoxazole, TET = Tetracycline, CAF = Chloramphenicol, CIP = Ciprofloxacin, GEN = Gentamycin

**Table 8. Drug resistance pattern and multi-drug resistance of MSSA and MRSA isolates for different antibiotics on nasal site among ART clients attending at Dessie comprehensive specialized hospital ART clinic from January to May 2020.**

| Isolated Micro-organisms | Number of Antibiotics with Resistance Strains (R%) | | | | | | | MDR strains (n = >R3) |
|---|---|---|---|---|---|---|---|---|
| | R1 | R2 | R3 | R4 | R5 | R6 | R7 | |
| *MSSA (n = 69)* | 34(49.3) | 21(30.4) | 6(8.7) | 8(11.6) | 0 | 0 | 0 | 14 (20.3) |
| MRSA (n = 58) | 0 | 2(3.4) | 5(8.6) | 19(32.8) | 17(29.3) | 13(22.4) | 2(3.4) | 56 (96.5) |
| Total (n = 127) | 34(26.8) | 23(18.1) | 11(8.7) | 27(21.3) | 17(13.4) | 13(10.2) | 2(1.6) | 70 (55.2) |

Key: R1, R2, R3, R4, R5, R6, and R7 = One, two, three, four, five, six and seven class of antibiotic resistance, respectively; MDR = Greater than three or more class of antibiotic resistance

In the present study *S. aureus* and MRSA isolates exhibit higher rate of sensitivity to gentamycin, clindamycin, chloramphenicol and ciprofloxacin. More or less the level of sensitivity for gentamycin, clindamycin, chloramphenicol and ciprofloxacin was in agreement with a studies conducted in Ethiopia [6, 23] and Ghana [14]. In the contrary, a study conducted in Nepal showed the majority of MRSA isolates was resistant to ciprofloxacin and gentamycin [8]. In the present study it was revealed that none of the isolates showed full level of sensitivity for the tested antibiotics. Whereas, few other similar studies indicated full sensitivity to the tested drugs [6, 17, 18]. This variation might be due to difference in the selection of antibiotics across different studies including the current one.

In the present study the whole isolates of *S. aureus* including MRSA showed full resistance for Penicillin which is in agreement with a study conducted in Uganda [18]. Like the present study another study [8] also indicated all MRSA isolate were found to be resistance to cefoxitin and penicillin. In other way the rate of MDR among *S. aureus* and MRSA isolates was found to be 55.2% and 96.5%, respectively. In comparison with previous studies [6, 27], the rate of MDR among MRSA isolates were found to be higher in the present study. This higher rate of antibiotics resistance and MDR might be partly due to the dissemination of antibiotic resistant *S. aureus* strain in the community and partly it might be due to utilization of antibiotics without antimicrobial susceptibility testing.

## Conclusions

In the present study a total of 206 HIV patients who attended ART clinic of the hospital were included, of which prominent proportion had *S. aureus* and MRSA nasal colonization. In the present study huge percentage of *S. aureus* and MRSA nasal colonization has noticed and this higher rate of colonization may have the chance of converting in to infection. Nasal colonization of *S. aureus* was significantly associated with educational status, having job close contact with human, most recent CD4 count, duration of initiation of HAART and line of ARV therapy. Whereas, MRSA colonization had statistically significant association with job close contact with human, habit of picking once nose and line of ARV therapy regiment. The antimicrobials sensitivity of both *S. aureus* and MRSA strain indicated an alarmingly increasing tendency of letting these population groups out of potential drug of choice for the treatment of infection due to the bacteria. Most MRSA but only one fifth MSSA strain had showed MDR. For these reasons health facility workers particularly ART personnel should prescribe antibiotics in accordance with drug susceptibility of the bacteria. Policy makers and medical directors of the health facilities should understand and manage MRSA among HIV patients and also stakeholders should find out the way how to decolonize the bacteria from nasal area. Proper health education should be also planned and given for HIV patients until they develop practical knowledge.

## Limitation of the study

Even though the present study provides excellent information about the nasal colonization of *S. aureus* and MRSA, there are still few limitations. First, the study was conducted in a single hospital that may not be representative of HIV patients who enrolled in the ART clinics of other health facilities. Second, it would be very good if the study included HIV non-infected populations as comparative group. Biofilm forming potential of MRSA and molecular detection of resistance were not executed.

## Acknowledgments

We deeply express our gratefulness to Wollo University and APHI-Dessie branch for all necessary supports without which this project would have not been possible. The authors would like to thank the study participants and their families. The authors also would like to acknowledge those health professionals who are working in ART clinic and laboratory personnel of Dessie comprehensive specialized hospital for their cooperation up to the accomplishment of this study.

## Author Contributions

**Conceptualization:** Hussein Muhaba, Daniel Gebretsadik.

**Data curation:** Hussein Muhaba.

**Formal analysis:** Hussein Muhaba, Genet Molla Fenta, Daniel Gebretsadik.

**Investigation:** Hussein Muhaba, Daniel Gebretsadik.

**Methodology:** Hussein Muhaba, Daniel Gebretsadik.

**Project administration:** Hussein Muhaba, Genet Molla Fenta, Daniel Gebretsadik.

**Resources:** Hussein Muhaba.

**Software:** Hussein Muhaba, Daniel Gebretsadik.

**Supervision:** Genet Molla Fenta, Daniel Gebretsadik.

**Validation:** Hussein Muhaba, Genet Molla Fenta.

**Visualization:** Genet Molla Fenta.

**Writing – original draft:** Hussein Muhaba, Daniel Gebretsadik.

**Writing – review & editing:** Hussein Muhaba, Genet Molla Fenta, Daniel Gebretsadik.

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
