## [Decision Letter · Decision Letter 0]

20 Apr 2022

PGPH-D-22-00354

Methicillin Resistance Staphylococcus aureus Nasal Carriage and Its Associated Factors Among HIV Patients Attending ART Clinic at Dessie Comprehensive Specialized Hospital, Dessie, North East Ethiopia

Dear %Daniel Gebretsadik%,

Thank you for submitting your manuscript to PLOS Global Public Health. After careful consideration, we feel that it has merit but does not fully meet PLOS Global Public Health’s publication criteria as it currently stands. Therefore, we invite you to submit a revised version of the manuscript that addresses the points raised during the review process.

Please submit your revised manuscript by . If you will need more time than this to complete your revisions, please reply to this message or contact the journal office at globalpubhealth@plos.org. Please include the following items when submitting your revised manuscript:

We look forward to receiving your revised manuscript.

Kind regards,

Abdullah Yusuf, M.Phil

Academic Editor

Journal Requirements:

1. Your co-authors, Hussein Muhaba (labhussein@yahoo.com) and Genet Molla Fenta (mgenetaskalu@gmail.com), have not confirmed authorship of the manuscript. We have resent them the authorship confirmation email; however please check that the above email address for them is correct and follow up personally to ensure they confirm. Please note that we cannot pass your manuscript to Production until we have received confirmations from all co-authors.

Just in case your co-authors are having difficulty confirming their authorship, you may advise them to send us an email at globalpubhealth@plos.org and we will confirm their authorship on the authors' behalf.

2. Please provide a detailed Financial Disclosure statement. This is published with the article, therefore should be completed in full sentences and contain the exact wording you wish to be published.

i) Please include all sources of funding (financial or material support) for your study. List the grants (with grant number) or organizations (with url) that supported your study, including funding received from your institution. 

ii). State the initials, alongside each funding source, of each author to receive each grant.

iii). State what role the funders took in the study. If the funders had no role in your study, please state: “The funders had no role in study design, data collection and analysis, decision to publish, or preparation of the manuscript.”

iv). If any authors received a salary from any of your funders, please state which authors and which funders.

3. Please update your Competing Interests statement. If you have no competing interests to declare, please state: “The authors have declared that no competing interests exist.”

4. In the online submission form, you indicated that “All Data are available up on proper request”. All PLOS journals now require all data underlying the findings described in their manuscript to be freely available to other researchers, either 1. In a public repository, 2. Within the manuscript itself, or 3. Uploaded as supplementary information.

5. We do not publish any copyright or trademark symbols that usually accompany proprietary names, eg (R), (C), or TM (e.g. next to drug or reagent names). Therefore please remove all instances of trademark/copyright symbols throughout the text, including Male®, 60®, Urban®, Single®, No®, 500®, First®, College and above®, No contact®, Undetected®, and ®; reference category.

Additional Editor Comments (if provided):

Minor Correction is needed. Please do the needful correction.

Reviewers' comments:

Reviewer's Responses to Questions

**Comments to the Author**

1. Does this manuscript meet PLOS Global Public Health’s publication criteria? Is the manuscript technically sound, and do the data support the conclusions? The manuscript must describe methodologically and ethically rigorous research with conclusions that are appropriately drawn based on the data presented.

Reviewer #1: Yes

Reviewer #2: Yes

2. Has the statistical analysis been performed appropriately and rigorously?

Reviewer #1: Yes

Reviewer #2: Yes

3. Have the authors made all data underlying the findings in their manuscript fully available (please refer to the Data Availability Statement at the start of the manuscript PDF file)?

Reviewer #1: Yes

Reviewer #2: Yes

4. Is the manuscript presented in an intelligible fashion and written in standard English?

Reviewer #1: Yes

Reviewer #2: Yes

5. Review Comments to the Author

Reviewer #1: Abstract: Factors related to the association of MRSA nasal carriage among HIV patients should be mentioned in the result section of the abstract.

Introduction: This section is nicely written. The association factor that has been written in the title was not described in the introduction. However, rationally should be written more precious and elaborately.

Methods: Sample Size determination(Line 101): please write sample size calculation in text form instead of formula.

Sample collection(Line 135): has been properly performed by the Triple package system (Line 139).

Culture (Line 156): Why Mac Conkey's media is used?

Ethics statement(Line 192): from where ethical clearance was taken? please mention it with the number.

Ethics approval (Line 375): This section is nicely written with mentions proper information.

Conclusion: Avoid Line 346 &347. It is repetition. The conclusion should be written in the present tense.

Table: Avoid the symbol of percentage % in the heading of the table. The author has said that in the result section Table 2 about CD4 count is baseline & recent. But this has not been mentioned in the methodology section. The procedure of measuring CD4 count is missing.

Table4: Does it duplicate with Table1?

Table5: Why age>60 years is considered as a reference?

Table5 & Table6:is it the repetition?

Table7: Mention either sensitive or resistant pattern. please avoid both.

Reviewer #2: The rate of S. aureus and MRSA nasal colonization was high and it has associated

43 with different factors. Understanding and managing MRSA among HIV patients is mandatory

44 and stakeholders should find out the way how to decolonize the bacteria from nasal area, SO it was a nice paper.

6. PLOS authors have the option to publish the peer review history of their article (what does this mean?). If published, this will include your full peer review and any attached files.

**Do you want your identity to be public for this peer review?** For information about this choice, including consent withdrawal, please see our Privacy Policy.

Reviewer #1: No

Reviewer #2: **Yes: **Arifa Akram

---

## [Editor Report · Decision Letter 1]

4 Jun 2022

PGPH-D-22-00354R1

Methicillin Resistance Staphylococcus aureus Nasal Carriage and Its Associated Factors Among HIV Patients Attending ART Clinic at Dessie Comprehensive Specialized Hospital, Dessie, North East Ethiopia

Dear Dr. Daniel Gebretsadik,

Thank you for submitting your manuscript to PLOS Global Public Health. After careful consideration, we feel that it has merit but does not fully meet PLOS Global Public Health’s publication criteria as it currently stands. Therefore, we invite you to submit a revised version of the manuscript that addresses the points raised during the review process.

We look forward to receiving your revised manuscript.

Kind regards,

M Abdullah Yusuf

Academic Editor

Journal Requirements:

Additional Editor Comments (if provided):

Dear Author,

Greetings!

Your paper has been reviewed and has found some minor corrections. Please do the needful correction and resubmit the paper.

Best wishes

Md Abdullah Yusuf
---

## [Editor Report · Decision Letter 2]

14 Jul 2022

PGPH-D-22-00354R2

Methicillin Resistance Staphylococcus aureus Nasal Carriage and Its Associated Factors Among HIV Patients Attending ART Clinic at Dessie Comprehensive Specialized Hospital, Dessie, North East Ethiopia

Dear Dr. Gebretsadik,

Thank you for submitting your manuscript to PLOS Global Public Health. After careful consideration, we feel that it has merit but does not fully meet PLOS Global Public Health’s publication criteria as it currently stands. Therefore, we invite you to submit a revised version of the manuscript that addresses the points raised during the review process.

We look forward to receiving your revised manuscript.

Kind regards,

Cemil Kurekci

Academic Editor

Journal Requirements:

Additional Editor Comments (if provided):

please explain what MDR is at first it is written.

"..Ethiopia reported a lower prevalence (6)" what is their findings?

For "Screening of Methicillin-Resistant S. aureus", did authors used positive and negative control strains. ?

Why authors did not use any molecular analysis to identify the strains as S. aureus.

Any control strains used for antimicrobial study?
---

## [Editor Report · Decision Letter 3]

29 Jul 2022

PGPH-D-22-00354R3

Methicillin Resistance Staphylococcus aureus Nasal Carriage and Its Associated Factors Among HIV Patients Attending ART Clinic at Dessie Comprehensive Specialized Hospital, Dessie, North East Ethiopia

Dear Dr. Gebretsadik,

Thank you for submitting your manuscript to PLOS Global Public Health. After careful consideration, we feel that it has merit but does not fully meet PLOS Global Public Health’s publication criteria as it currently stands. Therefore, we invite you to submit a revised version of the manuscript that addresses the points raised during the review process.

We look forward to receiving your revised manuscript.

Kind regards,

Cemil Kurekci

Academic Editor

Journal Requirements:

Additional Editor Comments (if provided):

Dear Author, thank you very much for your revision. I just would like to ask some further revisions, i) in material and method section, there is no need to write a quality control paragraph, so that just put control strain into necessary sections, and also add the ATCC number for strains (both resistant and sensitive).
---

## [Editor Report · Decision Letter 4]

11 Aug 2022

Methicillin Resistance Staphylococcus aureus Nasal Carriage and Its Associated Factors Among HIV Patients Attending ART Clinic at Dessie Comprehensive Specialized Hospital, Dessie, North East Ethiopia

PGPH-D-22-00354R4

Dear Mr. Gebretsadik,

We are pleased to inform you that your manuscript 'Methicillin Resistance Staphylococcus aureus Nasal Carriage and Its Associated Factors Among HIV Patients Attending ART Clinic at Dessie Comprehensive Specialized Hospital, Dessie, North East Ethiopia' has been provisionally accepted for publication in PLOS Global Public Health.

Best regards,

Cemil Kurekci

Academic Editor